# Unraveling the Molecular Mechanisms of Mosquito Salivary Proteins: New Frontiers in Disease Transmission and Control

**DOI:** 10.3390/biom15010082

**Published:** 2025-01-08

**Authors:** Jiayin Guo, Xiaoe He, Jianli Tao, Hui Sun, Jing Yang

**Affiliations:** 1Cuiying Biomedical Research Center, The Second Hospital & Clinical Medical School, Lanzhou University, Lanzhou 730030, China; 220220906411@lzu.edu.cn (J.G.); 220220906430@lzu.edu.cn (X.H.); sunhui@lzu.edu.cn (H.S.); 2Department of Pathology, Boston Children’s Hospital and Harvard Medical School, Boston, MA 02115, USA; jianli.tao@childrens.harvard.edu

**Keywords:** mosquito-borne disease, salivary protein, immune modulation, disease transmission

## Abstract

Mosquito-borne diseases are a group of illnesses caused by pathogens transmitted by mosquitoes, and they are globally prevalent, particularly in tropical and subtropical regions. Pathogen transmission occurs during mosquito blood feeding, a process in which mosquito saliva plays a crucial role. Mosquito saliva contains a variety of biologically active proteins that facilitate blood feeding by preventing blood clotting, promoting vasodilation, and modulating the host’s immune and inflammatory responses. These effects create an environment conducive to pathogen invasion and dissemination. Specific mosquito salivary proteins (MSPs) can promote pathogen transmission through mechanisms that either regulate hosts’ anti-infective immune responses or directly enhance pathogens’ activity. Strategies targeting these MSPs have emerged as an innovative and promising approach for the control of mosquito-borne diseases. Meanwhile, the diversity of these proteins and their complex interactions with the host immune system necessitate further research to develop safer and more effective interventions. This review examines the functional diversity of MSPs and their roles in disease transmission, discusses the advantages and challenges of strategies targeting these proteins, and explores potential future directions for research in this area.

## 1. Introduction

Mosquito-borne diseases, including malaria, dengue fever, West Nile fever, chikungunya fever, and Zika fever, are primarily transmitted by mosquitoes and collectively account for approximately one million deaths annually, posing a significant global public health burden [1,2]. Factors such as globalization, increased international travel, and changes in climate and land use patterns have heightened the risks of these diseases worldwide. Traditional control strategies have predominately relied on insecticides to target and kill mosquitoes; however, the widespread emergence of insecticide resistance in mosquito populations reduces their efficacy. Moreover, although vaccines such as Dengvaxia for dengue and RTS, S/AS01 and R21/Matrix-M for malaria are available, their efficacy remains limited in certain populations [3,4,5], and there is still an urgent need to develop more highly effective vaccines for the control of mosquito-borne diseases.

During blood feeding, infected female mosquitoes inject saliva containing pathogens into the skin of vertebrate hosts. Mosquito saliva contains numerous proteins with anticoagulant, vasodilatory, anti-inflammatory, and immunomodulatory properties [6,7]. These mosquito salivary proteins (MSPs) primarily facilitate blood feeding but also influence the transmission of diseases [8,9,10,11]. Among the estimated 100–200 proteins in mosquito saliva, 30–40% of them do not show similarity to any known protein, and their functions remain largely unknown [12]. Elucidating the molecular mechanisms by which MSPs influence disease transmission could provide valuable insights into novel prevention strategies. Moreover, MSPs could serve as biomarkers for the assessment of exposure to mosquito bites, enabling the estimation of disease transmission risks in specific regions [13,14]. Furthermore, vaccines targeting MSPs represent a promising approach to preventing certain mosquito-borne diseases. Overall, MSPs play an important role in the interactions among mosquitoes, pathogens, and vertebrate hosts, significantly influencing the occurrence and spread of these diseases [15,16].

## 2. Types of MSPs and Their Physiological Functions

Mosquito saliva contains a complex array of MSPs and other components [6,17,18]. Due to their anticoagulant, vasodilatory, immunoregulatory, and inflammation-modulating functions (Figure 1) [19], MSPs play a critical role in facilitating mosquito blood feeding and influencing pathogen transmission. The specific MSPs discussed in this review are summarized in Table 1.

### 2.1. Anticoagulation

Mosquitoes utilize their piercing–sucking mouthparts to extract blood for nutrition, but the host’s coagulation system acts as a defensive barrier, promoting blood clotting to prevent further bleeding. To overcome these hemostatic defenses, mosquitoes have evolved a repertoire of MSPs with anticoagulant functions [20].

Several MSPs exhibit anticoagulant activity by interfering with platelet function. For instance, Aegyptin, a protein from the saliva of *Aedes aegypti*, binds to collagen and inhibits platelet adhesion and aggregation, ultimately promoting successful blood feeding [21]. In line with this finding, a 30 kDa Aegyptin-like protein (alALP), identified from the salivary glands of female *Aedes albopictus*, has been shown to prolong the activated partial thromboplastin time (APTT), prothrombin time (PT), and thrombin time (TT) in vitro, as well as the bleeding time (BT) in vivo [22]. In addition, the anopheline antiplatelet protein (AAPP) isolated from the saliva of *Anopheles stephensi*—a human malaria vector in South Asia and the Persian Gulf—specifically inhibits collagen-induced platelet aggregation by binding to collagen [23]. Moreover, a 68 kDa recombinant *Ae*. *aegypti* salivary apyrase, an ecto-enzyme with ADPase and ATPase activity, inhibits human platelet aggregation induced by ADP, collagen, and thrombin [24].

MSPs can also disrupt blood coagulation by targeting specific clotting factors. Hamadarin from the saliva of *An*. *stephensi* suppresses the activation of the intrinsic coagulation pathway and subsequent release of bradykinin, a key mediator of inflammatory reactions. This inhibitory effect is due to hamadarin binding to coagulation factor XII (FXII) and high-molecular-weight kininogen (HMWK) [25]. The D7 proteins are among the most abundantly expressed proteins in the salivary glands of female mosquitoes. A recent study demonstrated that AngaD7L2, one of the three D7 long-form proteins expressed in the primary malaria vector *Anopheles gambiae*, exerts an anticoagulant effect by interacting with FXII, FXIIa, and FXI in the intrinsic coagulation pathway [26]. Additionally, a serine protease inhibitor found only in the female salivary glands of *Ae. aegypti* exhibits anticoagulant activity by inhibiting host FXa [27]. Moreover, anophelin is a 61-amino-acid peptide isolated from the salivary glands of *Anopheles albimanus*, a human malaria vector in Latin America. This peptide behaves as a tight-binding and specific inhibitor of thrombin [28,29]. Furthermore, a Kazal-type serine protease inhibitor, known as *Ae. aegypti* trypsin inhibitor (AaTI), is expressed in the female salivary glands. The recombinant AaTI protein has been shown to prolong the PT, APTT, and TT in vitro [30].

### 2.2. Vasodilation

Mosquito saliva contains vasodilatory substances that counteract the vasoconstriction triggered by mouthpart insertion during probing. This vasodilation increases the local blood flow, shortens the feeding time, and reduces the risk of host detection [31].

*Ae. aegypti* salivary glands produce sialokinins, which are vasodilatory peptides related to the tachykinin family. Early studies demonstrated that sialokinins exhibit vasodilatory effects comparable to those of the mammalian tachykinin substance P in vitro [32,33]. Recent research has further revealed the physiological relevance of sialokinins in blood feeding and host immune modulation by using *sialokinin*-knockout *Ae. aegypti* mosquitoes [34]. In the absence of sialokinins, mosquito bites fail to induce significant vasodilation, which is associated with longer probing times and reduced blood-feeding success on vertebrate hosts. Mechanistically, sialokinins induce vasodilation through the activation of nitric oxide synthase (NOS) via neurokinin-1 receptor (NK1R) signaling and disrupt the vasculature by enhancing the endothelial permeability. Additionally, mosquitoes lacking sialokinins show a diminished ability to recruit leukocytes and activate macrophages, which could influence pathogen transmission [34].

Anopheline mosquitoes do not produce vasodilatory substances but instead secrete MSPs that neutralize the host’s physiological vasoconstrictors or degrade vasoconstrictive amines. For instance, AngaD7L1 and AngaD7L3, the D7 long-form proteins in *An. gambiae*, bind to U-46619 (an analog of the potent vasoconstrictor thromboxane A2) and serotonin (a potent vasoconstrictor amine), respectively, inhibiting the vasoconstriction and platelet aggregation induced by these molecules [26]. In addition, a peroxidase/catechol oxidase is secreted by the salivary glands of female *An. albimanus* mosquitoes during probing. This enzyme exhibits vasodilatory activity by degrading vasoconstrictive amines, such as noradrenaline and serotonin [35,36].

Furthermore, MSPs can rapidly activate mast cells in the host’s skin by triggering a type I hypersensitivity reaction that results in the release of pro-inflammatory cytokines and histamine, which induces vasodilation [37,38].

### 2.3. Modulation of Inflammation and Immunity

Mosquito bites or saliva exert regulatory effects on the host’s inflammatory and immune responses, which can ultimately influence pathogen transmission [39,40,41,42,43,44,45,46,47,48,49,50,51,52,53].

Specific MSPs with immunomodulatory functions have been identified. For instance, a 34 kDa salivary protein from *Ae. aegypti*, namely neutrophil-stimulating factor 1 (NeSt1), binds strongly to human cluster of differentiation 47 (CD47). This interaction inhibits macrophage-mediated phagocytosis and dampens pro-inflammatory responses in white blood cells, thereby suppressing anti-Zika virus (ZIKV) responses in the skin [54]. Another example is salivary *Ae. aegypti* interleukin-4 (IL-4)-inducing protein (SAAG-4), which can program CD4^+^ T cells to express the signature Th2 cytokine IL-4 while reducing their production of the signature Th1 cytokine interferon-γ (IFN-γ) [55]. In addition, the salivary protein *An. gambiae* sporozoite-associated protein (AgSAP) was shown to bind to heparan sulfate and suppress local inflammatory responses in the skin [56].

More details on how specific MSPs regulate host immunity and influence the transmission of mosquito-borne pathogens will be provided in Section 3.

### 2.4. Other Functions

In addition to the above-mentioned roles, MSPs exhibit other functions. For example, serine protease activity detected in the saliva of *Ae. aegypti* enhances dengue virus (DENV) infectivity by proteolyzing extracellular matrix proteins, facilitating viral attachment and inducing cell migration. CLIP-domain serine protease A3 (CLIPA3), identified as a key salivary serine protease, plays a crucial role in this process [57].

An endonuclease has been identified in the saliva of female *Culex quinquefasciatus* mosquitoes. Although the relevance of endonuclease function in mosquito saliva is unclear, it may assist blood feeding by lowering the local viscosity at the bite site [58].

Research has also revealed that a putative antibacterial cecropin-like peptide (AAEL000598) found in *Ae*. *aegypti* saliva can kill various pathogens, including several Gram-negative bacteria, DENV, chikungunya virus (CHIKV), and *Leishmania* parasites [59].

Adenosine deaminase (ADA) hydrolyzes adenosine to inosine and ammonia. High ADA activity has been observed in the salivary glands of female *Cx. quinquefasciatus* and *Ae. aegypti* mosquitoes [60]. A possible role for salivary ADA is to reduce the local pain and itching caused by adenosine during mosquito feeding. Interestingly, recent studies have uncovered a novel role of ADA in virus replication within *Ae. aegypti* [61,62] and *Ae. albopictus* [63].

Over the past decade, advancements in vector biology have greatly expanded our understanding of MSPs. Nevertheless, many salivary proteins remain characterized, suggesting that mosquitoes may possess additional specialized activities that require further investigation.

### 2.5. Dynamic Changes in MSPs

The composition of MSPs is not static but undergoes dynamic changes [64]. Female mosquitoes primarily feed on plant sap when not ovipositing, but they switch to feeding on mammalian and avian blood during egg production [65]. These shifts in food sources are accompanied by corresponding changes in their salivary protein composition. Comparative proteomic studies of the salivary glands of sugar-fed and blood-fed *Ae*. *aegypti* females have shown that certain salivary proteins (e.g., anti-vasoconstrictive proteins) are overexpressed during blood feeding. These proteins help to block host responses to mosquito bites, facilitating successful blood feeding. This adaptability indicates that mosquitoes can modulate the composition of their salivary proteins based on their feeding requirements, ensuring that essential salivary proteins are expressed only when needed [66].

**Table 1 biomolecules-15-00082-t001:** The specific salivary proteins identified in mosquito vectors.

Category	Protein Name	Species	Function	Refs.
Anticoagulation	Aegyptin	*Ae. aegypti*	Binds collagen to block platelet aggregation	[21]
	alALP	*Ae. albopictus*	Prolongs APTT, PT, TT, and BT	[22]
	AAPP	*An. stephensi*	Binds collagen to block platelet aggregation	[23]
	Apyrase	*Ae. aegypti*	Hydrolyzes ADP to inhibit platelet aggregation	[24]
	Hamadarin	*An. stephensi*	Anticoagulation by inhibiting FXII	[25]
	AngaD7L2	*An. gambiae*	Interacts with FXII, FXIIa, and FXI to exert anticoagulant effect	[26]
	A serine protease inhibitor	*Ae. aegypti*	Anticoagulation by inhibiting FXa	[27]
	Anophelin	*An. albimanus*	Anticoagulation by inhibiting thrombin	[28,29]
	AaTI	*Ae. aegypti*	Prolongs PT, APTT, and TT	[30]
Vasodilation	Sialokinin	*Ae. aegypti*	Induces vasodilation by activating NK-1R signaling pathway to release NO	[32,33,34]
	AngaD7L1, AngaD7L3	*An. gambiae*	Scavenges vasoconstrictors to inhibit vasoconstriction	[26]
	Peroxidase/catechol oxidase	*An. albimanus*	Scavenges biogenic amines to induce vasodilation	[35,36]
Modulation of inflammation and immunity	SAAG-4	*Ae. aegypti*	Programs CD4^+^ T cells to express IL-4 and reduce IFN-γ production	[55]
Other functions	An endonuclease	*Cx. quinquefasciatus*	May lower local viscosity to assist blood feeding	[58]
	ADA	*Cx. quinquefasciatus, Ae. aegypti*	May reduce local pain and itching caused by adenosine	[60]
Malaria	AgTRIO	*An. gambiae*	Inhibits TNF-α expression; facilitates *Plasmodium* infection; enhances mosquito host-seeking behavior	[67,68,69]
	SAMSP-1	*An. gambiae*	Enhances sporozoite gliding and traversal abilities; facilitates *Plasmodium* infection	[70]
	AgSAP	*An. gambiae*	Binds to sporozoites and heparan sulfate; inhibits local skin inflammatory responses; facilitates *Plasmodium* infection	[56]
	mosGILT	*An. gambiae*	Suppresses sporozoite speed and cell traversal activity, lowering the initial parasite burden in mice	[71]
DENV	AT, ADA, 34-kDa protein, VA	*Ae. aegypti*	Promotes replication of DENV in human keratinocytes	[61]
	AaSG34	*Ae. aegypti*	Enhances DENV replication in mosquitoes and transmission in mice	[72]
	CLIPA3	*Ae. aegypti*	Hydrolyzes extracellular matrix proteins, increasing virus binding to heparan sulfate proteoglycans; induces cell migration; enhances DENV infectivity	[57]
	AaVA-1	*Ae. aegypti*	Activates autophagy in monocyte-derived cells, promoting dissemination of DENV in mice	[10]
	AaNRP	*Ae. aegypti*	Recruits neutrophils and other susceptible myeloid cells, promoting dissemination of DENV in mice	[73]
	A putative antibacterial cecropin-like peptide (AAEL000598)	*Ae. aegypti*	Inhibits DENV replication in C6/36 cells	[59]
	D7L1	*Ae. aegypti*	Binds to DENV virions; inhibits DENV infection in U937 cells and mice	[74]
	Aegyptin	*Ae. aegypti*	Increases the expression of GM-CSF, IFN-γ, IL-5, and IL-6; inhibits DENV infection in mice	[75]
WNV	AgBR1	*Ae. aegypti*	Enhances WNV pathogenicity in mice	[76]
CHIKV	A putative antibacterial cecropin-like peptide (AAEL000598)	*Ae. aegypti*	Inhibits CHIKV infection in HEK-293T cells	[59]
ZIKV	LTRIN	*Ae. aegypti*	Interferes with LTβR, blocking NF-κB signaling and pro-inflammatory cytokine production, thereby enhancing ZIKV pathogenicity in mice	[77]
	AgBR1	*Ae. aegypti*	Enhances ZIKV pathogenicity in mice	[78]
	NeSt1	*Ae. aegypti*	Suppresses local immune response, macrophage phagocytosis, and pro-inflammatory cytokine production, thereby enhancing ZIKV pathogenicity in mice	[54,79]
	AaVA-1	*Ae. aegypti*	Activates autophagy in monocyte-derived cells, promoting dissemination of ZIKV in mice	[10]
	AaNRP	*Ae. aegypti*	Recruits neutrophils and other susceptible myeloid cells, promoting dissemination of ZIKV in mice	[73]

alALP: *Ae. albopictus* Aegyptin-like protein; APTT: activated partial thromboplastin time; PT: prothrombin time; TT: thrombin time; BT: bleeding time; AAPP: anopheline anti-platelet protein; ADP: adenosine 5′-diphosphate; FXII: factor XII; AngaD7L2: *An. gambiae* D7 long-form protein 2; AaTI: *Ae. aegypti* trypsin inhibitor; NK-1R: neurokinin-1 receptor; NO: nitric oxide; SAAG-4: salivary *Ae. aegypti* IL-4-inducing protein; IL-4: interleukin-4; IFN-γ: interferon-γ; ADA: adenosine deaminase; AgTRIO: *An. gambiae* triple functional domain protein; TNF-α: tumor necrosis factor-α; SAMSP-1: sporozoite-associated mosquito saliva protein-1; AgSAP: *An. gambiae* sporozoite-associated protein; mosGILT: mosquito gamma-interferon-inducible lysosomal thiol reductase; DENV: dengue virus; AT: anti-thrombin; VA: venom allergen; AaSG34: *Ae. aegypti* salivary gland protein of 34 kDa; CLIPA3: CLIP-domain serine protease A3; AaVA-1: *Ae. aegypti* venom allergen-1; AaNRP: *Ae. aegypti* neutrophil recruitment protein; GM-CSF: granulocyte–macrophage colony-stimulating factor; WNV: West Nile virus; AgBR1: *Ae. aegypti* bacteria-responsive protein 1; CHIKV: chikungunya virus; ZIKV: Zika virus; LTRIN: lymphotoxin beta receptor inhibitor; LTβR: lymphotoxin-β receptor; NF-κB: nuclear factor kappa B; NeSt1: neutrophil-stimulating factor 1.

## 3. Effects of MSPs on Pathogen Infection and Disease Transmission

The occurrence, progression, and transmission of mosquito-borne diseases are complex processes. Initially, pathogens must infect mosquitoes, replicate within them, migrate to the salivary glands, and subsequently be transmitted to the mammalian host during the mosquito’s blood meal. Pathogens continuously cycle between mosquitoes and mammalian hosts, involving intricate interactions among the pathogen, mosquito, and host. This section focuses on how MSPs modulate these interactions, thereby affecting the development and transmission of various diseases. The mechanisms by which MSPs influence pathogen dissemination within the host are illustrated in Figure 2.

### 3.1. Malaria

Malaria is a global mosquito-borne disease caused by protozoan parasites of the genus *Plasmodium* and transmitted by *Anopheles* mosquitoes [80]. There are five major *Plasmodium* species that infect humans: *Plasmodium *falciparum**, *Plasmodium *vivax**, *Plasmodium *malariae**, *Plasmodium *ovale**, and *Plasmodium *knowlesi**. Among these, *P. falciparum* is the most prevalent and is responsible for the majority of severe cases and deaths, while *P. vivax* is notable for its widespread distribution and ability to cause relapsing infections via dormant liver-stage forms known as hypnozoites [81]. Common symptoms of malaria include fever, fatigue, vomiting, and headache; severe cases can result in jaundice, seizures, coma, and even death [1]. In 2022, malaria affected over 249 million people globally and caused 608,000 deaths, with more than 90% of cases occurring in Africa [82].

The lifecycle of *Plasmodium* parasites is complex, requiring both mosquitoes and humans. Within mosquitoes, *Plasmodium* undergoes gametogenesis and sporogony, classifying mosquitoes as the definitive host. In humans, *Plasmodium* undergoes schizogony and begins gametogenesis, making humans the intermediate host. Malaria transmission is influenced by climatic factors such as the temperature, humidity, and rainfall, which affect the lifecycles of both the mosquito vector and the parasite [83].

Previous studies have shown that the saliva or salivary gland extract (SGE) of *An. stephensi* can enhance the pathogenicity of *Plasmodium berghei* or *Plasmodium yoelii* in mice [84,85]. However, the natural immune response to mosquito bites does not afford protection against malaria in humans. Experimental animal models pre-exposed to mosquito bites or salivary gland components do not show consistent and substantial protection against *Plasmodium* infection [86,87,88,89]. In 2018, a study showed that antisera against *An. gambiae* SGE, prepared in rabbits using Freund’s adjuvant, could elicit robust and diverse responses to numerous proteins in mosquito saliva and confer partial protection against *Plasmodium* infection in mice [69]. Antibodies targeting one of the salivary antigens, *An. gambiae* triple functional domain protein (AgTRIO), contributed to this protective effect, suggesting that AgTRIO may serve as a vector-based target against malaria [69]. Moreover, subsequent studies using AgTRIO-deficient mosquitoes have revealed that AgTRIO influences *Plasmodium* transmission by modulating pro-inflammation cytokine expression at the bite site [67] and regulating mosquitoes’ probing capacity [68].

Another MSP, sporozoite-associated mosquito saliva protein-1 (SAMSP1), has been identified in *An. gambiae* and shown to facilitate malaria transmission. SAMSP1 enhances sporozoite gliding and cell traversal activity in vitro. Additionally, it inhibits neutrophil chemotaxis both in vivo and in vitro and increases the parasite burden in the liver of mice following the intradermal injection of *P. berghei*. Moreover, active or passive immunization with SAMSP1 reduces the liver parasite burden in mice infected with *P. berghei* through mosquito transmission [70].

In addition, *An. gambiae* sporozoite-associated protein (AgSAP), a protein derived from the saliva of *An. gambiae*, interacts with *P. falciparum* and *P. berghei* sporozoites without affecting their viability. AgSAP binds to heparan sulfate and modulates immune responses in the skin of mice. *AgSAP* knockout significantly reduces the ability of *Anopheles* mosquitoes to transmit *P. berghei* sporozoites to mice. Additionally, mice immunized with AgSAP show a reduced *Plasmodium* burden in the liver following infectious mosquito bites [56].

Furthermore, mosquito gamma-interferon-inducible thiol reductase (mosGILT), an MSP with inhibitory effects on the pathogenicity of *Plasmodium*, has also been identified in the saliva of infected female *Anophele* mosquitoes [71]. MosGILT binds to *Plasmodium* sporozoites and reduces their cell traversal activity in vitro and their motility in mouse skin. This inhibition leads to a reduced parasite burden in the liver of mice following the intradermal or intravenous injection of *P. berghei* sporozoites [71]. The exact mechanism by which mosGILT inhibits sporozoite activity remains to be determined, although it appears to depend on the C-terminal hydrophobic region of the protein [71]. Interestingly, subsequent research has shown that mosGILT is also expressed in mosquito reproductive systems and is required for their normal development. Female *An. gambiae* mosquitoes lacking mosGILT display underdeveloped ovaries incapable of producing eggs and show elevated thioester-containing protein 1 (TEP1)-dependent anti-*Plasmodium* innate immunity in the midgut [90]. Importantly, these findings suggest that MSPs like mosGILT could be multifunctional, with expression occurring at multiple stages and in various tissues. To better elucidate the specific roles of these MSPs in saliva and malaria transmission, a potential approach could involve generating conditional gene knockouts specifically in the mosquito salivary glands [91].

### 3.2. Dengue Fever

Dengue fever is a tropical mosquito-borne disease caused by DENV, primarily transmitted by *Aedes* mosquitoes, especially *Ae*. *aegypti* [92]. It remains the most prevalent arthropod-borne viral disease worldwide. In 2023, the World Health Organization (WHO) reported over 4.5 million cases and approximately 2300 deaths in the Americas alone [93]. Significant case numbers were also recorded in Asia, including Bangladesh (321,000 cases), Malaysia (111,400 cases), Thailand (150,000 cases), and Vietnam (369,000 cases) [93]. DENV is a single-stranded positive-sense RNA virus of the *Flavivirus* genus, with five distinct serotypes [94,95]. Infection with one DENV serotype confers lifelong immunity to that serotype but provides only short-term immunity to others. Subsequent infection with a different serotype increases the risk of severe complications due to antibody-dependent enhancement (ADE) [96,97]. Symptoms include high fever, headache, vomiting, muscle and joint pain, and characteristic skin rashes, with some cases progressing to life-threatening dengue hemorrhagic fever or dengue shock syndrome [92].

Several in vivo studies have demonstrated that *Ae. aegypti* mosquito bites and their SGE enhance DENV pathogenesis in the host [41,98,99]. However, a study showed that *Ae. aegypti* saliva inhibited DENV infection in human myeloid dendritic cells in vitro while increasing the secretion of IL-12p70 and tumor necrosis factor-α (TNF-α) in culture supernatants [100].

Specific *Ae. aegypti* MSPs are known to promote the pathogenesis of DENV. These include anti-thrombin (AT), an FXa-directed anticlotting serpin-like protein; ADA, a putative 34 kDa secreted salivary protein; and venom allergen (VA), a putative secreted protein, all of which enhance DENV replication in keratinocytes by inhibiting the expression of host antiviral immune genes [61]. In addition, a 34 kDa salivary protein of *Ae. aegypti* (AaSG34) is upregulated in the salivary glands following the ingestion of a DENV2-infected blood meal. Silencing *AaSG34* markedly reduces the DENV2 transcripts and envelope protein levels in the salivary glands after an infectious blood meal [72]. The intradermal inoculation of infectious mosquito saliva induced hemorrhaging in signal transducer and activator of transcription 1 (STAT1)-deficient mice, whereas saliva from *AaSG34*-silenced mosquitoes did not. These findings suggest that AaSG34 promotes DENV2 replication in the salivary glands and facilitates viral transmission [72]. Moreover, as previously mentioned, the *Ae. aegypti* salivary serine protease CLIPA3 enhances DENV infectivity by degrading extracellular matrix proteins. This suggests that serine protease inhibitors could be potential strategy to reduce DENV infection [57]. Furthermore, *Ae. aegypti* venom allergen-1 (AaVA-1) promotes the transmission of DENV and ZIKV in mice by activating autophagy in host monocyte lineage cells. Mechanistically, AaVA-1 interacts with leucine-rich pentatricopeptide repeat-containing protein (LRPPRC), a negative regulator of Beclin-1, which results in the release of Beclin-1 from LRPPRC-mediated sequestration and the initiation of autophagic signaling [10]. Another example is *Ae. aegypti* neutrophil recruitment protein (AaNRP). It enhances DENV and ZIKV transmission by promoting the rapid influx of neutrophils and the recruitment of virus-susceptible myeloid cells to bite sites. Mechanistically, AaNRP engages Toll-like receptor 1 (TLR1) and TLR4 on skin-resident macrophages, inducing the expression of neutrophil chemoattractants in a MyD88-dependent manner [73].

Interestingly, some MSPs have inhibitory effects on DENV pathogenesis. For instance, DENV infection upregulates a gene encoding a cecropin-like peptide (AAEL000598), a small cationic antimicrobial peptide found in the salivary glands of female *Ae. aegypti*. The recombinant form of this peptide effectively inhibits DENV replication in *Ae. albopictus* C6/36 cells in a dose-dependent manner [59]. The D7 protein, another saliva component of *Ae. aegypti*, inhibits DENV infection in the human macrophage cell line U937 and reduces viral infection in mice, likely through its direct interaction with DENV virions [74]. Aegyptin, a salivary protein known for its anticoagulant properties, is less abundant in the saliva of *Ae. aegypti* mosquitoes infected with DENV. Compared to mice inoculated with DENV alone, those co-inoculated with Aegyptin and DENV exhibit reduced DENV titers at the inoculation site and in circulation, along with increased levels of granulocyte–macrophage colony-stimulating factor (GM-CSF), IFN-γ, IL-5, and IL-6. These findings suggest that Aegyptin provides negative pressure on viral perpetuation. This pressure may be inherent to the Aegyptin protein family and perhaps impacts viral transmission [75].

### 3.3. West Nile Fever

West Nile fever is caused by the West Nile virus (WNV), primarily transmitted by *Culex* mosquitoes, although it has also been isolated from *Ae*. *aegypti* [101,102]. Originating in Africa, WNV has now spread globally, with large-scale outbreaks reported in Europe and North America in recent decades [103]. WNV is a single-stranded, positive-sense RNA virus of the *Flavivirus* genus. While most human infections are asymptomatic, WNV can cause a wide range of symptoms, from mild fever to severe encephalitis [104].

Bites or SGE from *Culex tarsalis* and *Ae. aegypti* exacerbate WNV infection in mice, regardless of the infection route, whether via mosquito transmission, subcutaneous injection, or intradermal inoculation [48,105,106,107]. A previous study showed that prior exposure to *Ae. aegypti* bites increases mortality in mice infected with WNV via mosquito transmission [108]. In contrast, intramuscular immunization with *Cx. tarsalis* SGE and a synthetic peptide adjuvant reduces mortality in similarly infected mice [109]. These results mirror the differential protective effects observed between mosquito bites and SGE immunization in *Anopheles*-borne malaria [69]. The immune activation induced by natural mosquito bites differs fundamentally from that elicited by artificial hyper-immunization with SGE, potentially leading to varied impacts on disease transmission and progression.

Additionally, in vitro experiments showed that saliva from *Cx. quinquefasciatus* or *Ae. aegypti* suppresses the inflammatory response in primary human keratinocytes infected with WNV, but only *Ae. aegypti* saliva modulates WNV replication [110].

Moreover, *Ae. aegypti* bacteria-responsive protein 1 (AgBR1) has been identified as a protein upregulated in the salivary glands following blood feeding. Treatment with antibodies targeting AgBR1 effectively reduces the initial viral load and delays lethal infection in mice infected with WNV via mosquito bites, suggesting that AgBR1 could be a potential therapeutic target [76].

### 3.4. Chikungunya Fever

Chikungunya fever is caused by CHIKV, primarily transmitted by *Ae*. *aegypti* and *Ae. albopictus* mosquitoes [111,112]. CHIKV belongs to the *Togaviridae* family and the *Alphavirus* genus, with its genome composed of a single-stranded positive-sense RNA molecule [113]. In recent years, chikungunya fever has spread rapidly worldwide, with Brazil being the most affected country in the Americas, reporting over 1.6 million cases [114]. It has become a significant public health issue in these regions. The main symptoms include fever, joint pain, nausea, vomiting, rash, and myalgia. While the symptoms typically resolve within a week, some patients may experience prolonged joint pain [115]. Patients with comorbidities, specific genetic conditions, or an advanced age are at greater risk for severe disease and increased mortality [116].

In vitro studies have shown that the saliva and SGE of *Aedes* mosquitoes suppress the antiviral responses of infected cells and promote CHIKV replication [117,118]. Similarly, *Aedes* mosquito saliva can suppress the antiviral response in mouse skin, thereby enhancing CHIKV replication and disease progression [40,45].

Despite the overall proviral effects of saliva and SGE, a peptide with antiviral activity has been identified in the salivary glands of *Aedes* mosquitoes. As mentioned in the previous section, this cecropin-like peptide inhibits the CHIKV infection of the human HEK-293T cell line in a dose-dependent manner [59].

### 3.5. Zika Fever

Zika fever is a zoonotic disease caused by ZIKV, which is primarily transmitted to humans through *Ae*. *aegypti* mosquitoes [119]. Recent studies have also indicated that ZIKV can be transmitted sexually between humans and can be passed from mother to fetus [120,121,122]. ZIKV was first isolated from the blood of rhesus macaques and from *Aedes* mosquitoes in Africa in the mid-20th century [123]. Since then, ZIKV has been confirmed globally, with different strains isolated across continents. In the past decade, outbreaks of ZIKV infections in different regions have attracted significant attention from the scientific community [124,125]. ZIKV is a single-stranded positive-sense RNA flavivirus. The primary symptoms of ZIKV infection include rash, fever, headache, dizziness, fatigue, anorexia, and abdominal pain, which are similar to those of other viral infections [126]. However, a notable feature of ZIKV infection is its association with neurological complications in both newborns and adults [126,127].

In vitro experiments demonstrate that *Ae. aegypti* SGE suppresses DENV- and ZIKV-induced inflammasome activation in human and mouse macrophages, while mitigating virus-induced cell death without altering viral replication [128]. Currently, several specific MSPs from *Ae. aegypti* have been identified that promote ZIKV infection, replication, dissemination, and transmission by modulating the host’s immune response. After blood feeding, female *Ae. aegypti* upregulate the expression of a 15 kDa protein named lymphotoxin beta receptor inhibitor (LTRIN) in their salivary glands. This protein preferentially inhibits nuclear factor kappa B (NF-κB) signaling and the production of inflammatory cytokines by interfering with the dimerization of lymphotoxin-β receptor (LTβR), thereby enhancing ZIKV infection in cells in vitro and exacerbating its pathogenesis in mice in vivo. Notably, mice treated with anti-LTRIN antibodies exhibited resistance to mosquito-mediated ZIKV infection [77]. Another protein, AgBR1, in the saliva of *Ae. aegypti* promotes the inflammatory response of murine splenocytes in vitro and exacerbates ZIKV infection and disease in vivo. Antiserum against AgBR1 partially protects mice from lethal mosquito-borne ZIKV infection [78]. NeSt1 from *Ae. aegypti* activates primary mouse neutrophils ex vivo and alters the immune environment at the mosquito bite site in vivo, enhancing ZIKV replication and dissemination during the early stages of infection [79]. Nest1 can also suppress phagocytosis by human macrophages and inhibit pro-inflammatory responses in white blood cells through its interaction with CD47. This interaction subsequently reduces antiviral responses and promotes ZIKV dissemination in human skin explants [54]. The aforementioned AaVA-1 from *Ae. aegypti* not only promotes the transmission of DENV in mice by activating autophagy in host monocyte lineage cells but also facilitates the transmission of ZIKV in mice through the same mechanism [10]. As previously mentioned, AaNRP from female *Ae. aegypti* also promotes the transmission of ZIKV. Additionally, dietary supplementation with resveratrol, an anti-inflammatory phytochemical, was found to reduce the influx of cutaneous neutrophils mediated by mosquito saliva, thereby suppressing ZIKV transmission [73].

Some salivary proteins from female *Ae. aegypti* can also directly interact with ZIKV. For example, AAEL000793, AAEL007420, and AAEL006347 bind to the envelope protein of ZIKV with nanomolar affinities. However, this interaction does not affect the replication of ZIKV in human endothelial cells and keratinocytes in vitro [129].

### 3.6. Other Viral Infections

Aside from the aforementioned viruses, there is a variety of other viruses transmitted by mosquitoes. However, many of these viruses are much less studied, and there is a lack of research on the roles of specific MSPs in regulating the infection and transmission of these viruses. For example, the co-injection of Rift Valley fever virus (RVFV) with SGE or saliva from *Aedes* mosquitoes via the intradermal route increased mortality in mice and elevated the viral titers in multiple organs and the blood [130]. *Ae. aegypti* bites can increase the replication and dissemination of Semliki Forest virus (SFV) and Bunyamwera virus in mice injected with these viruses, leading to higher mortality rates [44]. Interestingly, the co-injection of Japanese encephalitis virus (JEV) and SGE collected from *Cx. quinquefasciatus* into domestic pigs resulted in milder febrile illness and a shortened duration of nasal shedding, but did not have a measurable effect on viremia or neuroinvasion [131].

In addition, the saliva or bites of *An. gambiae* and *Ae. aegypti* did not affect the viral loads in mouse tissues or serum viral titers following needle inoculation with O’nyong-nyong virus (ONNV) [132]. This suggests that the pathogenicity and transmissibility of ONNV may rely more on its own characteristics rather than modulation by mosquito saliva.

## 4. Surveillance Strategies for Mosquito-Borne Diseases by Targeting MSPs

Currently, the most commonly used indicator to measure the transmission risk of mosquito-borne diseases is the entomological inoculation rate (EIR) [133]. The EIR, a metric to estimate the number of bites by infectious mosquitoes per person per unit time, is obtained via the human landing catch (HLC) [134]. Nevertheless, this approach is constrained by several factors, including significant resource usage, limited measurement sensitivity, and ethical issues [135]. Additionally, it only provides a rough estimate of the mosquito exposure at a specific location [136]. Recent years have seen the development of surveillance strategies for mosquito-borne diseases based on mosquito salivary biomarkers (SBs). This method assesses the transmission risk by detecting antibodies in human serum that target specific MSPs. Several studies have shown that this approach provides more reliable data and holds promise as an effective tool for the evaluation of the disease transmission risk and the success of control measures [66,137,138,139,140,141,142,143].

Several *Anopheles* salivary proteins have been shown to be effective in monitoring exposure to *Anopheles* bites. The IgG antibody response to the salivary proteins gSG6 and cE5 in *An. gambiae* serves as an indicator of human exposure to *Anopheles* bites. The two proteins elicit distinct immune responses, suggesting that their applicability as SBs may differ [144,145]. The IgG response to *Anopheles* salivary protein D7 is consistent with the known intensity of malaria transmission in different regions. Notably, the IgG response against the long-form D7 protein, D7L2, increases with age and is lower in individuals who use insecticide-treated bednets, highlighting its potential as an SB for the evaluation of human–vector exposure and in assessing the impact of vector control measures [13]. Additionally, *An. gambiae* salivary proteins AgSAP and AgTRIO have been identified as reliable indicators of recent exposure to bites from mosquitoes infected with *P. falciparum* in low- and moderate-transmission areas [14].

Several *Aedes* salivary proteins have also been identified as potential SBs. In dengue-endemic areas, the human antibody response to *Ae. aegypti* D7 salivary proteins is associated with age, living conditions, and DENV infection, suggesting that D7 proteins have the potential to serve as SB candidates [146]. A study conducted in children found that IgG responses against the long-form D7 proteins from *Ae. aegypti*, namely D7L1 and D7L2, can serve as a sensitive and highly specific method to assess human exposure to *Ae. aegypti* bites, with no cross-reactivity to other mosquito species. Nevertheless, this study was limited to children, and additional data are needed to further validate the use of this dual-protein-based evaluation approach in other populations, particularly in adults [147]. Furthermore, the correlation between anti-AgBR1 and anti-NeSt1 IgG levels and factors such as dengue severity, symptom duration, and hospital disease management indicates that these MSPs could be important tools in determining prior exposure to DENV and evaluating the risk of dengue severity progression [148]. Bites from *Ae. albopictus* or *Ae. aegypti* trigger species-specific IgG responses to the salivary proteins al34k2 and ae34k2, respectively, in mice. However, the researchers validated the immunogenicity of al34k2 and its limited immune cross-reactivity with ae34k2 using only a single human serum sample exhibiting hyperimmunity to *Ae. albopictus* saliva [149]. Measurements of the humoral response in larger groups of individuals naturally exposed to *Aedes* bites will be necessary to further evaluate the potential of these antigens in detecting human exposure to *Ae. albopictus* and *Ae. aegypti*. Additionally, there is a strong positive correlation between the specific human IgG response to the *Aedes* N-term 34 kDa salivary peptide (derived from the *Ae. aegypti*’s 34k1 salivary protein) and the *Aedes* density, which is influenced by individual characteristics, climatic factors, and vector control interventions. However, this response has not been shown to be associated with the dengue transmission risk [150].

There are currently some challenges associated with the application of SBs. Recombinant proteins may carry multiple epitopes, which can increase the risk of immune cross-reactivity, thus reducing the specificity of candidate SBs [137]. A peptide-based approach has been developed and holds promise in improving the detection specificity [137]. There is currently a lack of specific SBs designed to assess bites from infectious mosquitoes, which would allow for a direct evaluation of the corresponding infectious disease transmission risk [14,151]. Additionally, the detection of SBs requires specialized personnel, which limits their widespread use. The development of rapid diagnostic kits will be crucial for the broader adoption of SB-based detection methods in the future. In conclusion, mosquito SBs are an emerging tool for the assessment of mosquito-borne disease risks, offering a safer, more accurate, and more efficient alternative to traditional methods. By combining the strengths of traditional assessment methods with those of mosquito SB-based measurements, a more comprehensive and accurate evaluation of mosquito-borne disease transmission risks and the effectiveness of public health interventions can be achieved.

## 5. Recent Advances in Vaccines Targeting MSPs

Approved vaccines targeting mosquito-borne diseases include the yellow fever vaccine YF-VAX, the malaria vaccines RTS, S/AS01 and R21/Matrix-M, the dengue vaccine Dengvaxia (chimeric yellow fever–dengue tetravalent vaccine), and the chikungunya vaccine Ixchiq [4,152,153,154,155]. There are no licensed vaccines for ZIKV or WNV to date. The approved vaccines mentioned above primarily target pathogens [156]. While they have provided some protection in endemic areas, their limited effectiveness highlights the need for safer and more effective vaccines. Vaccines targeting MSPs may offer a novel approach to preventing mosquito-borne diseases, providing broad-spectrum protection against multiple pathogens transmitted by specific mosquito species [157].

To date, AGS-v, a vaccine containing four synthetic salivary peptides derived from *An. gambiae* salivary proteins, including salivary gland 7-like protein, salivary gland 1-like protein (from which two peptides were derived), and gSG7 protein, has completed a phase 1 clinical trial to assess its safety and immunogenicity in humans. The results showed that AGS-v was well tolerated and immunogenic [158]. AGS-v PLUS, an advanced version of AGS-v incorporating a fifth peptide antigen named AGS-20, which is found in the saliva of many mosquito species, demonstrated a similar safety and immunogenicity profile in humans [159]. These findings suggest that vaccination targeting MSPs is safe and holds potential as a feasible strategy to reduce the burden of mosquito-borne diseases. Additional studies are necessary to assess the protective efficacy of these vaccines against mosquito-borne diseases in humans.

In addition to AGS-v, various MSPs have been studied as potential vaccine targets in experimental models for the prevention of mosquito-borne diseases. Among these, AgTRIO has demonstrated potential as a vaccine target. Both the active immunization of mice with AgTRIO and passive immunization with AgTRIO antiserum have been shown to provide a degree of protection to mice subsequently infected with *P. berghei.* The authors also conducted experiments using *P. falciparum* and humanized mice for passive immunization, demonstrating that the above findings can be extended to the human pathogen. Moreover, AgTRIO antiserum and an antibody targeting the circumsporozoite protein (CSP), a key surface protein of malaria parasites in the sporozoite stage, work synergistically to provide protection [69]. Additionally, passive immunization with AgBR1 antiserum delayed lethal infections in mice bitten by ZIKV- or WNV-infected *Ae. aegypti* [76,78]. Meanwhile, studies have demonstrated that the active immunization of mice with AgBR1, adjuvanted with aluminum hydroxide, delayed lethal mosquito-borne ZIKV infection, highlighting the potential of AgBR1 as a vaccine component to combat ZIKV [160]. Both AgBR1 and NeSt1 antisera, when used individually, provided partial protection against ZIKV infection in mice, each altering the early host response in the skin and reducing viremia [78,79]. Moreover, combining these antisera enhanced survival and reduced the viral burden, offering more effective protection than either antiserum alone. This suggests that a combined immunization strategy targeting multiple MSPs, such as AgBR1 and NeSt1, could provide a more effective approach to preventing mosquito-borne ZIKV infection [161]. Nonetheless, MSP-based vaccines carry certain risks. Studies have shown that administering a vaccine containing the recombinant salivary protein D7 from *Cx. tarsalis*, followed by the mosquito transmission of WNV, resulted in more severe disease and higher mortality rates in mice. Furthermore, transferring serum from vaccinated mice to naïve mice led to similarly severe mosquito-transmitted WNV disease, suggesting that the anti-D7 antibodies induced by the vaccine contributed to the enhanced severity of the disease [162]. This emphasizes the critical importance of selecting vaccine targets carefully to ensure both safety and efficacy.

Currently, many studies are using immunoinformatics approaches to design multi-epitope subunit vaccines targeting MSPs. The process involves epitope incorporation, 3D modeling and refinement, molecular docking with immune receptors, and in silico cloning into expression vectors [163,164,165]. Such vaccines, designed using immunoinformatics, offer advantages such as high immunogenicity, non-allergenicity, broad coverage, and structural stability. Nonetheless, further clinical validation is needed to confirm their safety, immunogenicity, and efficacy for practical applications.

Research on vaccines targeting MSPs has made significant progress but also faces some challenges. First, due to the complexity of mosquito saliva, identifying key targets requires extensive fundamental research and screening efforts. Second, the efficacy of such vaccines may be influenced by individual differences and environmental factors, necessitating validation in a wide range of populations to confirm their general applicability [166]. Moreover, some salivary proteins may vary across different mosquito species, complicating vaccine design and implementation [157]. Despite these challenges, vaccines targeting MSPs have tremendous potential. Their unique mechanism of action not only offers a novel approach for the control of mosquito-borne diseases but also paves the way for new research directions in vaccine development. By deepening our understanding of the functions and mechanisms of MSPs, scientists can design safer and more effective vaccines, contributing significantly to global mosquito-borne disease control efforts.

## 6. Conclusions and Remarks

Mosquito-borne diseases impose a significant burden on global health, necessitating the urgent exploration of novel prevention and control strategies. Research on MSPs has made remarkable progress, revealing their roles in anticoagulation, vasodilation, and the regulation of hosts’ immune and inflammatory responses. These functions create favorable conditions for the invasion and dissemination of pathogens. Specific MSPs can promote pathogen transmission by modulating the host’s immune response or directly enhancing pathogens’ activity. Targeted vaccines against these specific MSPs hold promise in effectively interrupting the transmission of corresponding mosquito-borne diseases. However, rigorous clinical trials are essential to evaluate and monitor the safety and effectiveness of such vaccines. Additionally, some MSPs exhibit immunogenicity, inducing a strong IgG antibody response in the host. Surveillance strategies that target these MSPs as SBs offer a promising approach to assessing the transmission intensity of mosquito-borne diseases.

Future research should focus on identifying more novel MSPs and further elucidating their specific roles and mechanisms in disease transmission. Since the expression of MSPs varies among mosquito species and across different physiological states, comparative analyses should be conducted to gain a comprehensive understanding of their diversity and dynamic properties. Furthermore, integrating disciplines such as structural biology and bioinformatics will enable systematic studies of MSPs and expand their applications in the prevention and control of mosquito-borne diseases.

Although significant progress has been made in the study of MSPs over the past few decades, many questions still remain unanswered. With advancements in technology and innovative research methods, our understanding of these proteins is expected to deepen. In conclusion, research on the role of MSPs in disease transmission holds not only great academic value but also profound implications for public health. We look forward to witnessing more groundbreaking discoveries in the future, which will drive innovation and improvements in strategies for the prevention and control of mosquito-borne diseases.

## Figures and Tables

**Figure 1 biomolecules-15-00082-f001:**
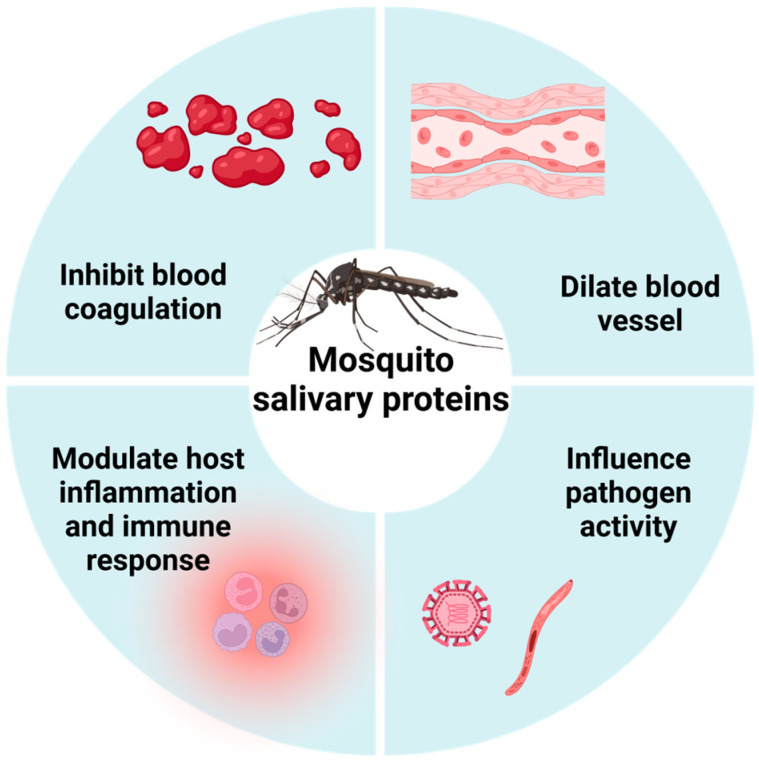
Schematic representation of the functions of MSPs. Created in BioRender. Guo, J. (2024).

**Figure 2 biomolecules-15-00082-f002:**
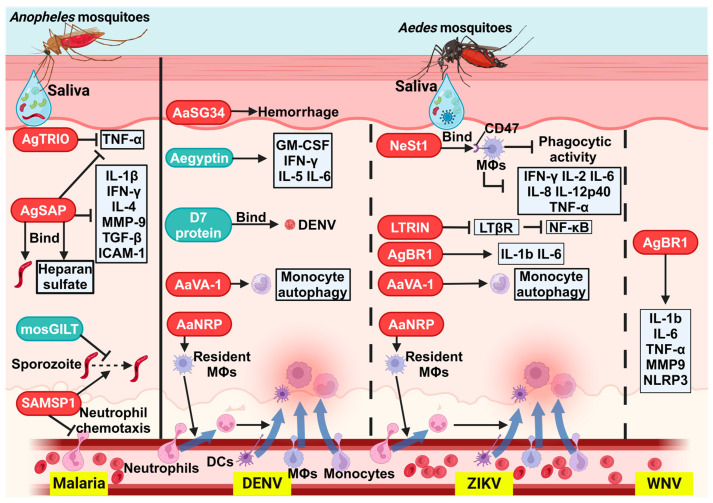
Schematic illustration of the mechanisms by which specific MSPs influence pathogen infection and transmission. The red oval frames represent proteins that promote pathogen infection in the host, while the blue oval frames indicate those that inhibit it. AgTRIO: *An. gambiae* TRIO; AgSAP: *An. gambiae* sporozoite-associated protein; mosGILT: mosquito gamma-interferon-inducible lysosomal thiol reductase; SAMSP-1: sporozoite-associated mosquito saliva protein-1; AaSG34: *Ae. aegypti* salivary gland protein of 34 kDa; AaVA-1: *Ae. aegypti* venom allergen-1; AaNRP: *Ae. aegypti* neutrophil recruitment protein; NeSt1: neutrophil-stimulating factor 1; LTRIN: lymphotoxin beta receptor inhibitor; AgBR1: *Ae. aegypti* bacteria-responsive protein 1; DCs: dendritic cells; MΦs: macrophages; TNF-α: tumor necrosis factor-α; IL-4: interleukin-4; MMP-9: matrix metalloproteinase-9; TGF-β: transforming growth factor-β; ICAM-1: intercellular adhesion molecule-1; GM-CSF: granulocyte–macrophage colony-stimulating factor; IFN-γ: interferon-γ; LTβR: lymphotoxin-β receptor; NF-κB: nuclear factor kappa B; NLRP3: NOD-like receptor family pyrin domain-containing 3; DENV: dengue virus; ZIKV: Zika virus; WNV: West Nile virus. Created in BioRender. Guo, J. (2024).

## Data Availability

Not applicable.

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
