# Peer review of "Unraveling the Molecular Mechanisms of Mosquito Salivary Proteins: New Frontiers in Disease Transmission and Control"

_biomolecules, 2025, doi:10.3390/biom15010082_

Round 1
Reviewer 1 Report
Comments and Suggestions for Authors
General comments
The manuscript reviews research on mosquito salivary proteins and their function in obtaining an efficient bloodmeal and in vectoring pathogens. There is a lot of good information here, but I is very long and often the different sections do read a bit like a shopping list of published papers with very little contextual interpretation of these studies that would be expected in a review piece. For example, the term “Another study” occurs 13 times as does the term“A study” occurs 12 times. I think more work on the writing would improve this review in better connecting the pieces of information in some sections. There is a lot of text on vaccines for pathogens that can be removed.
Specific comments
L43: please provide a reference for this statement and sentence…” Moreover, MSPs 43 can serve as biomarkers to assess exposure to mosquito bites, thus estimating disease transmission risks in specific areas.”
L79 – remove cap from Aegypti
Table 1 could be formatted more efficiency and the bottom legend the species names don’t require qualifications as that has been done earlier in the text.
L218: please cite this statement “In 2020, over 240 million people worldwide were infected with malaria, resulting in 620,000 deaths, with more than 90% of cases occurring in sub-Saharan Africa”
L302 This statement requires a citation “Another study indicated that the salivary protein AaSG34 from Ae. aegypti is crucial for DENV-2 replication in mosquitoes and transmission in mice. When the gene encoding AaSG34 was knocked out, DENV genomic RNA and envelope protein transcripts were significantly reduced in the salivary glands of infected mosquitoes.”
L512: This sentence is unclear “For instance, indoor HLC cannot accurately estimate outdoor biting, thus failing to provide a comprehensive assessment of overall vector exposure”
L490: the section on biomarkers is currently called “ Control strategies for mosquito-borne diseases targeting MSPs”. I think it would be better served being changed to “Surveillance strategies…” And the section on availability of vaccines for vector pathogens us unnecessary in this review and would place the section on salivary peptide vaccines as a separate section.
Comments on the Quality of English LanguageThe writing is OK but the structure and the sentence and paragraph writing can be improved. Paragraphs are often too long and and have too many points within them, which often do no link together. When citing studies, it can be useful to link relates studies that use similar molecular mechanisms.
Author Response
Comments 1: L43: please provide a reference for this statement and sentence…”Moreover, MSPs can serve as biomarkers to assess exposure to mosquito bites, thus estimating disease transmission risks in specific areas.”
Response 1: Thank you for pointing this out. We have added the relevant references to support the statement and please refer to references [13, 14], line 48.
Comments 2: L79 – remove cap from Aegypti
Response 2: Thank you for your suggestion. We have replaced "Aegypti" with "aegypti" (Line 66).
Comments 3: Table 1 could be formatted more efficiency and the bottom legend the species names don’t require qualifications as that has been done earlier in the text.
Response 3: Thank you for your valuable feedback. We have reformatted Table 1 to improve its efficiency by removing the qualifications for species names in the bottom legend (Please refer to Pages 5-7).
Comments 4: L218: please cite this statement “In 2020, over 240 million people worldwide were infected with malaria, resulting in 620,000 deaths, with more than 90% of cases occurring in sub-Saharan Africa”
Response 4: Thank you for your suggestion. We have added the appropriate reference to support the statement. The reference is from the World Health Organization's World Malaria Report 2023 (please refer to reference [83], line 209).
Comments 5: L302 This statement requires a citation “Another study indicated that the salivary protein AaSG34 from Ae. aegypti is crucial for DENV-2 replication in mosquitoes and transmission in mice. When the gene encoding AaSG34 was knocked out, DENV genomic RNA and envelope protein transcripts were significantly reduced in the salivary glands of infected mosquitoes.”
Response 5: Thank you for pointing this out. We have added the appropriate reference to support the statement (Please refer to reference [73], lines 303-306).
Comments 6: L512: This sentence is unclear “For instance, indoor HLC cannot accurately estimate outdoor biting, thus failing to provide a comprehensive assessment of overall vector exposure”
Response 6: Thank you for your feedback. We have removed the sentence following the restructuring of the manuscript to improve clarity and coherence.
Comments 7: L490: the section on biomarkers is currently called “ Control strategies for mosquito-borne diseases targeting MSPs”. I think it would be better served being changed to “Surveillance strategies…” And the section on availability of vaccines for vector pathogens us unnecessary in this review and would place the section on salivary peptide vaccines as a separate section.
Response 7: Thank you for your valuable suggestion. We have revised the section title to “Surveillance strategies for mosquito-borne diseases targeting MSPs” as recommended (Please refer to line 445). Additionally, we have created a separate section focusing on vaccines targeting MSPs, with only a brief introduction of currently available vaccines for vector pathogen (Please refer to lines 507-511).
Comments 8: Comments on the Quality of English Language
The writing is OK but the structure and the sentence and paragraph writing can be improved. Paragraphs are often too long and and have too many points within them, which often do no link together. When citing studies, it can be useful to link relates studies that use similar molecular mechanisms.
Response 8: Thank you for your valuable and constructive feedback. We have revised the structure and improved the sentence and paragraph organization to ensure better flow and coherence. Long paragraphs have been divided into shorter, focused sections to address specific points more clearly. Additionally, We have linked related studies, particularly those utilizing similar molecular mechanisms, to enhance the narrative and provide a more comprehensive discussion. We truly appreciate your insightful suggestions, which have significantly improved the quality of the manuscript.
Reviewer 2 Report
Comments and Suggestions for Authors
COMMENTS:
This is an interesting review of salivary proteins of vector mosquitoes, providing valuable insights into their role in disease with description for improving our understanding of vector-borne disease transmission, as alternative for development of novel vector control strategies, such as vaccine development and as biomarker development. However, there is some points that can be improve.
This is an interesting review of salivary proteins in vector mosquitoes, providing valuable insights into their role in disease transmission. It also discusses their potential as targets for novel vector control strategies, including vaccine development and the creation of biomarkers. However, there are a few areas that could be improved."
Comments and suggestions below:
1. Line 31: this reference only discusses malaria, dengue and West Nile virus. What about chikungunya and Zika virus?
2. Lines 35 and 36: I suggest briefly describing the Dengue and Malaria vaccines, even you provide a sentence about Vaccine.
3. Lines 45 to 48: Your description is about mosquitoes, but reference 7 is about sand flies.
4. Lines 57 to 60: add reference for this paragraph.
5. Line 61: When you mention a scientific name for the first time, do not abbreviate the genus in parentheses. However, in subsequent references to the same species, you have used the genus abbreviation. Please check this throughout the manuscript.
6. Line 64: in vitro instead in vitro. Please check this throughout the manuscript.
7. Lines 75 to 76: The scientific name should be in italics. Please check this throughout the manuscript and reference list.
Another observation is regarding the phrase “….., a human malaria vector….”. This information should be providing the first time you describe An. stephensi (Line 68).
8. Line 86: in vivo instead in vivo. Please check this throughout the manuscript.
9. Lines 88 to 92: Add reference.
10. Lines 93 to 95: Add reference.
11. Lines 106 to 107: “Modulating host inflammation and immune response” and “Influencing pathogen transmission” are not the same thing. Could you explain the difference?
12. Lines 110 to 112: Add reference.
13. Line 142: Do not abbreviate the scientific name at the beginning of the sentence. Please check this throughout the manuscript.
14. Lines 148 to 149: Reference 42 does not support what is described in this sentence.
15. Lines 153 to 155: Please provide a reference for chikungunya.
16. Lines 176 to 180: Reference 55 does not support what is describe in this sentence.
17. Line 191: Add reference for this sentence: The composition of MSPs is not static but undergoes dynamic changes.
18. Line 191 to 192: Add reference for this information: Female mosquitoes primarily feed on plant sap when not ovipositing,…
19. Line 201: Table 1, column “Protein names”, check the size letter for all protein names.
- I did not find this information in reference 21
- I suggest placing all on the same line for refence 65.
- Please, double-check this information in reference 92.
- Table legend: Scientific name should not include in the legend; this is a universal nomenclature. However, there are many abbreviations (for protein names) that were not included in the legend.
20. Lines 218 to 221: Please, provide more details about malaria, similar to how you described arbovirus. For example, mention that malaria is cause by protozoa, the species that affect humans and another relevant information.
21. Lines 221 to 223: There is more recent data; and please add the reference.
22. Lines 270 to 272: Do you have more recent data?
23. Lines 383 to 384: this reference https://pubmed.ncbi.nlm.nih.gov/25674945/ is more appropriate that 101.
24. Line 459: Ae. aegypti instead Aedes aegypti.
25. Lines 490 to 492: Please provide references for two sentences.
26. Lines 499 to 506: Please provide references for the sentences.
27. Line 523: Use the abbreviation, the reference about this abbreviation was provided in the line 501.
- Anopheles instead Anopheles.
28. Line 551: Aedes instead Aedes.
29. Lines 597 to 605: Please provide reference for the sentences.
30. Line 597: I suggest starting with the Yellow fever vaccine, which is older and highly effective, and then continue with malaria and dengue vaccines, which are more recent.
Another point is: what about Zika and Chikungunya? Even though we don't have vaccines for them, it's important to mention these diseases. Are there any studies on them?
31. Line 644: An. gambiae instead Anopheles gambiae.
31. Lines 747: Check all scientific names, they should write in italic.
Author Response
Comments 1: Line 31: this reference only discusses malaria, dengue and West Nile virus. What about chikungunya and Zika virus?
Response 1: Thank you for the comment. We have added a relevant reference discussing chikungunya and Zika virus (Please refer to reference [2], line 28.)
Comments 2: Lines 35 and 36: I suggest briefly describing the Dengue and Malaria vaccines, even you provide a sentence about Vaccine.
Response 2: Thank you for your valuable comment. We have added a sentence briefly describing the vaccines for dengue and malaria (Please refer to lines 35-39).
Comments 3: Lines 45 to 48: Your description is about mosquitoes, but reference 7 is about sand flies.
Response 3: Thank you for your comment. We apologize for the oversight and we have removed the reference related to sand flies.
Comments 4: Lines 57 to 60: add reference for this paragraph.
Response 4: Thanks for your suggestion. We have added reference [20] to support the paragraph (Please refer to lines 61-64).
Comments 5: Line 61: When you mention a scientific name for the first time, do not abbreviate the genus in parentheses. However, in subsequent references to the same species, you have used the genus abbreviation. Please check this throughout the manuscript.
Response 5: Thank you for your insightful comment. We have carefully reviewed the manuscript and ensured that the full genus name is used when a scientific name is mentioned for the first time, and the genus abbreviation is used for subsequent references. We appreciate your attention to detail, which has helped improve the consistency of the manuscript.
Comments 6: Line 64: in vitro instead in vitro. Please check this throughout the manuscript.
Response 6: Thank you for pointing out this. We have thoroughly reviewed the manuscript and corrected the formatting of in vitro throughout the text.
Comments 7: Lines 75 to 76: The scientific name should be in italics. Please check this throughout the manuscript and reference list.
Another observation is regarding the phrase “….., a human malaria vector….”. This information should be providing the first time you describe An. stephensi (Line 68).
Response 7: Thank you for your valuable comments. We have thoroughly reviewed the manuscript and reference list to ensure that all scientific names are correctly italicized. Additionally, we have relocated the phrase “a human malaria vector” to the first mention of An. stephensi as suggested. (Please refer to line 72).
Comments 8: Line 86: in vivo instead in vivo. Please check this throughout the manuscript.
Response 8: Thank you for pointing this out. We have carefully reviewed the manuscript and ensured the correct formatting of in vivo throughout the text.
Comments 9: Lines 88 to 92: Add reference.
Response 9: Thank you for the comment. We have added reference [31] to support the content (Please refer to Line 95).
Comments 10: Lines 93 to 95: Add reference.
Response 10: As you suggested, we have added the relevant reference [34] to support the content (Please refer to Line 101).
Comments 11: Lines 106 to 107: “Modulating host inflammation and immune response” and “Influencing pathogen transmission” are not the same thing. Could you explain the difference?
Response 11: Thank you for your insightful comment. We think that "modulating host inflammation and immune response" and "influencing pathogen transmission" are distinct processes, although they may overlap in some aspects. To clarify:
Modulating the host's inflammation and immune response refers to the ability of mosquito saliva to suppress or alter the host's immune reaction. This can affect pathogen transmission. For example, the salivary protein AaNRP from Aedes aegypti can rapidly and efficiently recruit neutrophils and other mosquito-borne virus-susceptible myeloid cells, facilitating the dissemination of viruses within the host.
In addition to mosquito salivary proteins effects on the host, they might also directly target the pathogen, thereby influencing its transmission. For instance, the salivary protein SAMSP-1 from Anopheles gambiae can directly enhance the gliding and traversal ability of Plasmodium sporozoites, promoting the infection of the host by the malaria parasite.
To avoid ambiguity, we would like to change the label in Figure 1 from 'Influence pathogen transmission' to 'Influence pathogen activity.
We hope that our explanation meets your requirement.
Comments 12: Lines 110 to 112: Add reference.
Response 12: We have added the appropriate reference [39-53] to support the content (Please refer to Line 128).
Comments 13: Line 142: Do not abbreviate the scientific name at the beginning of the sentence. Please check this throughout the manuscript.
Response 13: Thanks for your careful examination. We have carefully reviewed the manuscript and ensured that scientific names are not abbreviated at the beginning of sentences.
Comments 14: Lines 148 to 149: Reference 42 does not support what is described in this sentence.
Response 14: We apologize for the oversight. We have removed the sentence following the restructuring of the manuscript to improve clarity and coherence. We appreciate your careful review, which has helped improve the accuracy of the manuscript.
Comments 15: Lines 153 to 155: Please provide a reference for chikungunya.
Response 15: Thank you for your comment. The sentence has been removed due to the restructuring of the manuscript.
Comments 16: Lines 176 to 180: Reference 55 does not support what is describe in this sentence.
Response 16: Thank you for your comment and we apologize for the oversight. We have rephrased the statement and added an appropriate reference to support this point (Please refer to lines 156-161). We believe this revision improves clarity and accuracy.
Comments 17: Line 191: Add reference for this sentence: The composition of MSPs is not static but undergoes dynamic changes.
Response 17: We have added the appropriate reference [64] to support the statement (Please refer to Line 168).
Comments 18: Line 191 to 192: Add reference for this information: Female mosquitoes primarily feed on plant sap when not ovipositing,…
Response 18: As you suggested, we have added the appropriate reference [65] to support the statement (Please refer to Line 170).
Comments 19: Line 201: Table 1, column “Protein names”, check the size letter for all protein names.
- I did not find this information in reference 21
- I suggest placing all on the same line for refence 65.
- Please, double-check this information in reference 92.
- Table legend: Scientific name should not include in the legend; this is a universal nomenclature. However, there are many abbreviations (for protein names) that were not included in the legend.
Response 19: Thank you for your detailed and valuable comments. We sincerely apologize for the oversight and have carefully reviewed and addressed each point (Please refer to Pages 5-6):
The size and formatting of all protein names in the “Protein names” column have been checked and corrected.
Errors in reference citations have been corrected or removed as necessary.
The table legend has been revised to remove scientific names, as they are universal nomenclature. Additionally, abbreviations for protein names have been added to the legend to improve clarity.
Comments 20: Lines 218 to 221: Please, provide more details about malaria, similar to how you described arbovirus. For example, mention that malaria is cause by protozoa, the species that affect humans and another relevant information.
Response 20: Thank you for your valuable comment. We have expanded the section on malaria to provide additional details, including its cause by protozoan parasites of the genus Plasmodium, the species that infect humans, and other relevant information, such as the lifecycle of Plasmodium and transmission factors. We believe these additions provide a more comprehensive overview of malaria, similar to the level of detail used in the arbovirus section. Please refer to lines (202-207) and lines (212-217).
Comments 21: Lines 221 to 223: There is more recent data; and please add the reference.
Response 21: Thank you for your suggestion. We have rephrased the statement by citing more recent data, along with the supporting reference (Please refer to lines 209-211).
Comments 22: Lines 270 to 272: Do you have more recent data?
Response 22: Thank you for the comment. We have rephrased the sentence by citing more recent data, along with the supporting reference (Please refer to lines 283-286).
Comments 23: Lines 383 to 384: this reference https://pubmed.ncbi.nlm.nih.gov/25674945/ is more appropriate that 101.
Response 23: Thank you for your insightful suggestion regarding the reference. We have placed your recommended reference in the content (please refer to reference [113], line 365).
Comments 24: Line 459: Ae. aegypti instead Aedes aegypti.
Response 24: We have made the corresponding changes (Please refer to line 420).
Comments 25: Lines 490 to 492: Please provide references for two sentences.
Response 25: Thank you for your comment. The sentence has been removed due to the restructuring of the manuscript.
Comments 26: Lines 499 to 506: Please provide references for the sentences.
Response 26: Thank you for the comment. We have added the appropriate references [134-137] (Please refer to lines 446-452).
Comments 27: Line 523: Use the abbreviation, the reference about this abbreviation was provided in the line 501.
- Anopheles instead Anopheles.
Response 27: Thank you for the comment. We have replaced the full term with the abbreviation as suggested. (Please refer to line 447).
Additionally, we have italicized Anopheles as suggested.
Comments 28: Line 551: Aedes instead Aedes.
Response 28: We have corrected the word as suggested (Please refer to line 470).
Comments 29: Lines 597 to 605: Please provide reference for the sentences.
Response 29: We have added the appropriate references [4] [153-156] (Please refer to lines 507-511).
Comments 30: Line 597: I suggest starting with the Yellow fever vaccine, which is older and highly effective, and then continue with malaria and dengue vaccines, which are more recent.
Another point is: what about Zika and Chikungunya? Even though we don't have vaccines for them, it's important to mention these diseases. Are there any studies on them?
Response 30: Thank you for your thoughtful comment. We have revised this section to begin with the Yellow Fever vaccine, followed by a discussion of the more recent malaria and dengue vaccines. Regarding Chikungunya, we noted the recent development and approval of a vaccine, such as the FDA-approved Ixchiq vaccine, which represents significant progress in combating this virus. For Zika and West Nile viruses, we acknowledged the current lack of available vaccines. (Please refer to lines 507-511).
Comments 31: Line 644: An. gambiae instead Anopheles gambiae.
Response 31: We have made the corresponding revision (please refer to line 517).
Comments 32: Lines 747: Check all scientific names, they should write in italic.
Response 32: Thank you for the comment. We have carefully reviewed and ensured that all scientific names are written in italics as required. We appreciate your thorough review and attention to detail.